# FrCas9 is a CRISPR/Cas9 system with high editing efficiency and fidelity

Zifeng Cui[1,7], Rui Tian[2,3,7], Zhaoyue Huang[1,7], Zhuang Jin[1], Lifang Li[1], Jiashuo Liu[1], Zheying Huang[1], Hongxian Xie[4], Dan Liu[4], Haiyan Mo[4], Rong Zhou[4], Bin Lang[5], Bo Meng[6], Haiyan Weng[6✉] & Zheng Hu [1,3✉]

Genome editing technologies hold tremendous potential in biomedical research and drug development. Therefore, it is imperative to discover gene editing tools with superior cutting efficiency, good fidelity, and fewer genomic restrictions. Here, we report a CRISPR/Cas9 from *Faecalibaculum rodentium*, which is characterized by a simple PAM (5′-NNTA-3′) and a guide RNA length of 21–22 bp. We find that FrCas9 could achieve comparable efficiency and specificity to SpCas9. Interestingly, the PAM of FrCas9 presents a palindromic sequence, which greatly expands its targeting scope. Due to the PAM sequence, FrCas9 possesses double editing-windows for base editor and could directly target the TATA-box in eukaryotic promoters for TATA-box related diseases. Together, our results broaden the understanding of CRISPR/Cas-mediated genome engineering and establish FrCas9 as a safe and efficient platform for wide applications in research, biotechnology and therapeutics.

[1] Department of Gynecological oncology, the First Affiliated Hospital, Sun Yat-sen University, Zhongshan 2nd Road, Yuexiu, Guangzhou 510080 Guangdong, China. [2] Center for Translational Medicine, The First Affiliated Hospital, Sun Yat-sen University, Guangzhou, 510080 Guangdong, China. [3] Sun Yat-sen University Nanchang Research Institution, Nanchang 330200 Jiangxi, China. [4] Generulor Company Bio-X Lab, Zhuhai 519000 Guangdong, China. [5] School of Health Sciences and Sports, Macao Polytechnic Institute, Macao 999078, China. [6] Department of Pathology, The First Affiliated Hospital of USTC, Division of Life Sciences and Medicine, University of Science and Technology of China, Hefei 230001 Anhui, China. [7] These authors contributed equally: Zifeng Cui, Rui Tian, Zhaoyue Huang. ✉email: whaiyan1166@163.com; huzheng1998@163.com

Repurposing the prokaryotic immune system CRISPR-Cas for genome engineering has revolutionized the field of biological technology, enabling extensive applications in scientific research, biomedicine and agriculture[1]. Among many CRISPR proteins, SpCas9 with the NGG PAM is the most widely-used gene editing tool due to (i) it is the first CRISPR system investigated and reported, and more importantly, (ii) it is considered to possess the highest editing efficiency, which is the key factor of effective gene modifications[2,3]. However, accompanied by high cutting efficiency, SpCas9 also generates off-targets, which occur in genomic sites resembling the target sequences[4]. These unwanted side-effects may cause catastrophe effects in the genome and severely hinder the utility of SpCas9 in basic and clinical applications[4]. Several strategies have been developed to reduce the off-target effects. The first is to develop algorithms based on massive off-target data to select the optimal sgRNAs but may restrict the targetable genomic loci[5,6]. The second is to engineer SpCas9 protein functional domains to improve its specificity but may reduce the efficiency of the nuclease[7–9]. The third is to engineer guide RNA, including truncated gRNAs[8], extended gRNAs[10] and chemically modified gRNAs[11].

To identify CRISPR proteins with advantages in both specificity and size, more orthologs were reported, including Type II-A Cas9 (*Staphylococcus aureus*, SaCas9), Type II-C Cas9 (*Neisseria meningitidis*, NmeCas9; *Campylobacter jejuni*, CjCas9; *Geobacillus stearothermophilus*, GeoCas9; *Pasteurella pneumotropica*, PpCas9)[12–15] and Type V Cpf1[16–19]. However, these CRISPR nucleases showed weakened cleavage activity, making them less competitive for highly efficiency-required applications. Therefore, to find CRISPR systems with comparable efficiency as SpCas9, at the same time possessing improved specificity and distinct PAM requirements is in urgent need.

In this study, a Type II-A FrCas9, derived from *Faecalibaculum rodentium*[20], is identified with distinct biochemical characteristics from reported CRISPR systems. We find that FrCas9 can efficiently edit target DNA sequences with a 5′-NNTA-3′ PAM in both prokaryotic and eukaryotic genomes. The palindromic PAM increases the densities of sgRNA distributions in various organisms and has advantage in targeting TATA box within eukaryotic promoters. Further, we show that FrCas9 could achieve comparable efficiency and specificity to SpCas9, providing a powerful platform for biomedical research and drug development.

## Results

**FrCas9 edits distinct 5′-NNTA-3′ PAM.** By bioinformatic screening, we identified the Type II-A system in the genomes of *Faecalibaculum rodentium*. The phylogenetic analysis showed that FrCas9 is dissimilar to SpCas9 at a distance of 1.80 (Fig. 1a and Supplementary Table 1), indicating it may possess different characteristics. The FrCas9 locus included a CRISPR array composed of 31 spacer-direct repeat units (Supplementary Table 2), locating adjacent *to cas1, cas2, csn2* and *cas9* genes (Fig. 1b). By searching sequences complementary to the direct repeats[21], we identified a 71 nt tracrRNA sequence (not included the poly T) upstream the *cas9* gene (Fig. 1b and Supplementary Table 3). Next, we predicted 7 catalytic residues in FrCas9 HNH and RuvC domains (Fig. 1b and Supplementary Table 4).

To confirm the prediction of tracrRNA, we validated its expression from small RNA-seq of *E. coli* that contained simplified FrCas9 locus. The tracrRNA sequences were actively transcribed in an opposite orientation to the *cas* genes, while the crRNA sequences were transcribed in the same direction as the *cas* genes (Fig. 1b). Further analysis indicated that the spacers and direct repeats of mature crRNA products mainly distributed in 21-22 nt and 20 nt, respectively (Fig. 1c, d).

Next, we conducted a depletion assay to determine PAM requirement of FrCas9[22]. A plasmid library containing a 30 bp target site following 6 bp random PAM sequences was constructed and electroporated into *E. coli* containing FrCas9 plasmid (Fig. 1e and Supplementary Table 5). Only the targets meeting PAM requirements would be depleted from *E. coli*. The weblogo and PAM wheel showed that FrCas9 may have potential 5′-NRTA-3′ PAM (Fig. 1e and Supplementary Fig. 1, N: A, T, C, G; R: A, G). Repeated depletion assay (n = 4) revealed that the 2nd PAM position had slight preference for G (mean log2 fold-change = 0.21) or A (mean log2 fold-change = 0.11) (Fig. 1f). Then, we specifically validated the 2nd PAM preference by the plasmid interference assay[23]. Compared to control vectors, four 5′-NNTA-3′ groups all displayed inhibited cell growth under the dual-antibiotic screening (Fig. 1g and h), indicating that there was no 2nd preference in the PAM of FrCas9. Together, FrCas9 edited DNA sequences with a 5′-NNTA-3′ PAM in prokaryotic cells.

**FrCas9 is active in eukaryotic cells.** To test the genome editing activity of FrCas9 in mammalian cells, we joined the whole 71 nt tracrRNA and the 42 nt crRNA as a single guide RNA (sgRNA) by GAAA-tetraloop[3]. Then, above sgRNA and synthesized human codon optimized FrCas9 sequences were cloned into the PX330 plasmid vector.

We developed the puromycin depletion assay to confirm the PAM sequence of FrCas9 in HEK293T cell lines (Fig. 2a and Supplementary Table 5). As a positive control, the 5′-NGG-3′ PAM of SpCa9 validated our workflow (Fig. 2b). FrCas9 had an obvious preference for 3rd T (Log2 fold-change = 0.17) and 4th A (Log2 fold-change = 0.23), validating the 5′-NNTA-3′ PAM requirements in prokaryotic cells (Fig. 2c). To further characterize the PAM preference of FrCas9 in living human cells, we conducted high-throughput PAM determination assay (HT-PAMDA)[24], which showed that FrCas9 had a canonical 5′-NNTA-3′ PAM with scattered NNTG, NNAN and NNGT non-canonical PAMs (Supplementary Fig. 2b).

To further confirm the editing rates on 5′-NNTA-3′ PAM sequences in human cells, we first tested the genome editing ability of FrCas9 targeting 12 sites in 3 human endogenous genes (Supplementary Table 5). The 12 sites included all four nucleotides in the 2nd position of PAMs (NGTA, NATA, NCTA, NTTA). The TIDE assay showed that all sites with cleavage activities to various extents (Fig. 2d). Further, we validated the 5′-NNTA-3′ editing events at additional 32 sites in 8 gene loci, indicating the FrCas9 had efficient cleavage on 5′-NNTA-3′ PAM sequences in human cells (Supplementary Fig. 3). After the double-stranded breaks (DSBs), Cas9s commonly result in the non-homologous end joining (NHEJ) repair events, for instance, incorporating end-protected double-stranded oligonucleotides (dsODNs), which was used in GUIDE-seq[4]. We further detected the incorporation of dsODNs of 4 sgRNAs targeting the *RNF2* gene through dsODN-PCR[25], and successfully verified the FrCas9 on-target modification (Fig. 2e and Supplementary Fig. 4). Therefore, FrCas9 required a 5′-NNTA-3′ PAM in eukaryotic cells.

**The optimal sgRNA designs for FrCas9.** Previous studies showed that the sgRNA consisting of full-length crRNA and tracrRNA may not achieve the best editing efficiency[5,26]. Therefore, we further investigated the optimal sgRNA architecture of FrCas9 by truncating both the 3′ terminal of crRNA (referred as repeat:antirepeat truncation) and tracrRNA (referred as tracrRNA truncation) (Fig. 2f). The ability of the truncated sgRNAs to generate indels in human *RNF2* gene (Supplementary Table 5) was tested and compared with SpCas9[5]. Similar to other Cas9 orthologs[5,15], our data showed that the truncations of 3′

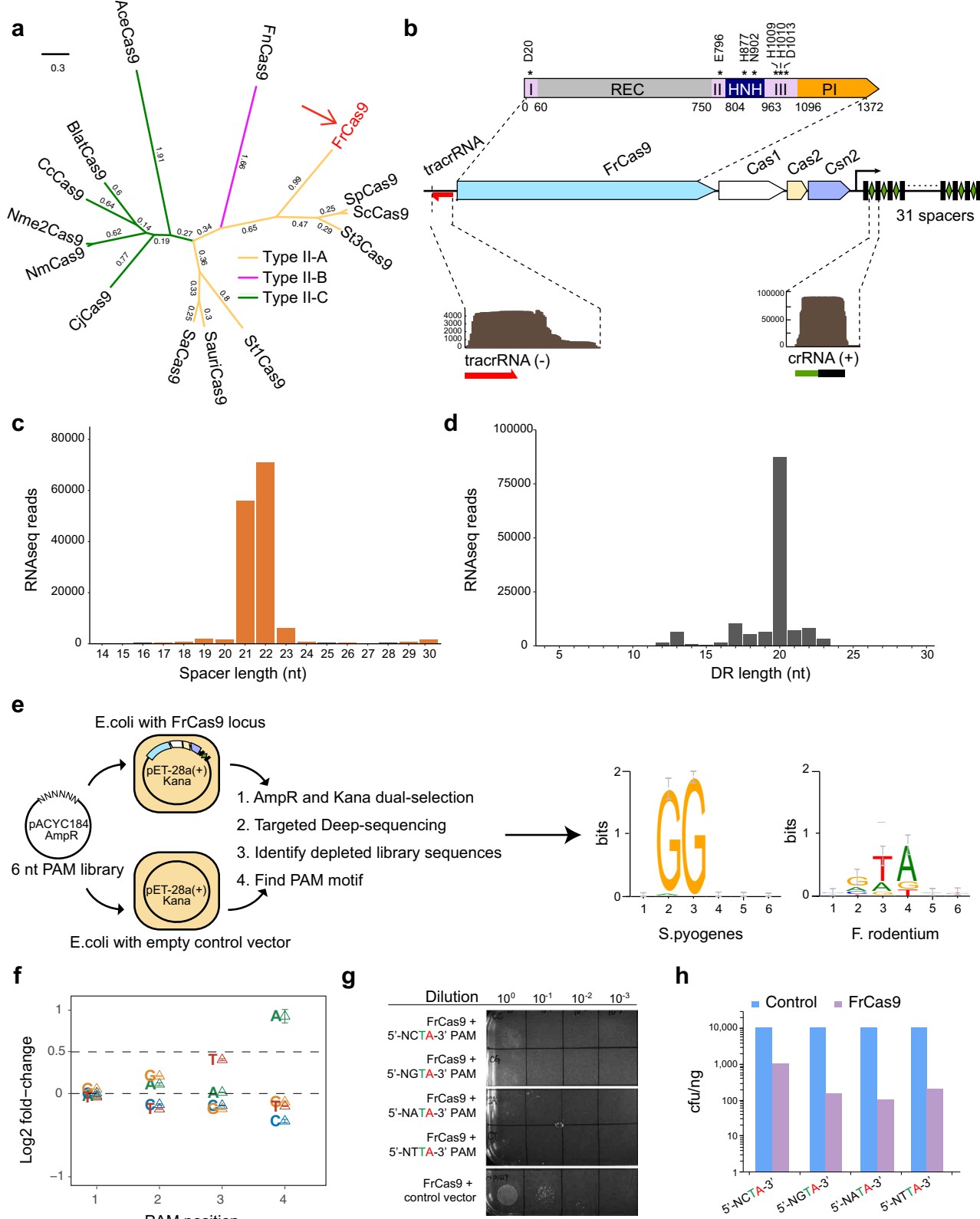

**Fig. 1 FrCas9 edits distinct 5'-NNTA-3' PAMs. a** The phylogenetic tree of FrCas9 and 13 active Cas9 orthologs. **b** The schematic of *Faecalibaculum rodentium* systems. Insert above displayed the domains of FrCas9 with active residues indicated with asterisks. I, II, III represented three RuvC domains. Insert below showed expressed crRNA and tracrRNA from small RNA-seq of *E. coli* harboring pET-28A plasmid with simplified FrCas9 locus. **c** Distribution of the length of spacer sequences derived from small RNA sequencing. **d** Distribution of the length of DR sequences derived from small RNA sequencing. **e** The schematic of depletion assay and web-logo results for SpCas9 and FrCas9. **f** The deletion effects of first four nucleotides of FrCas9 PAM. Data are presented as mean values ± S.E.M. (*n* = 4). **g** The plasmid interference assay of FrCas9 in 4 sites that differed in the 2nd PAM positions. A series of dilution was performed. **h** The bar plot of cell units of above plasmid interference assay. Source data are provided with this paper.

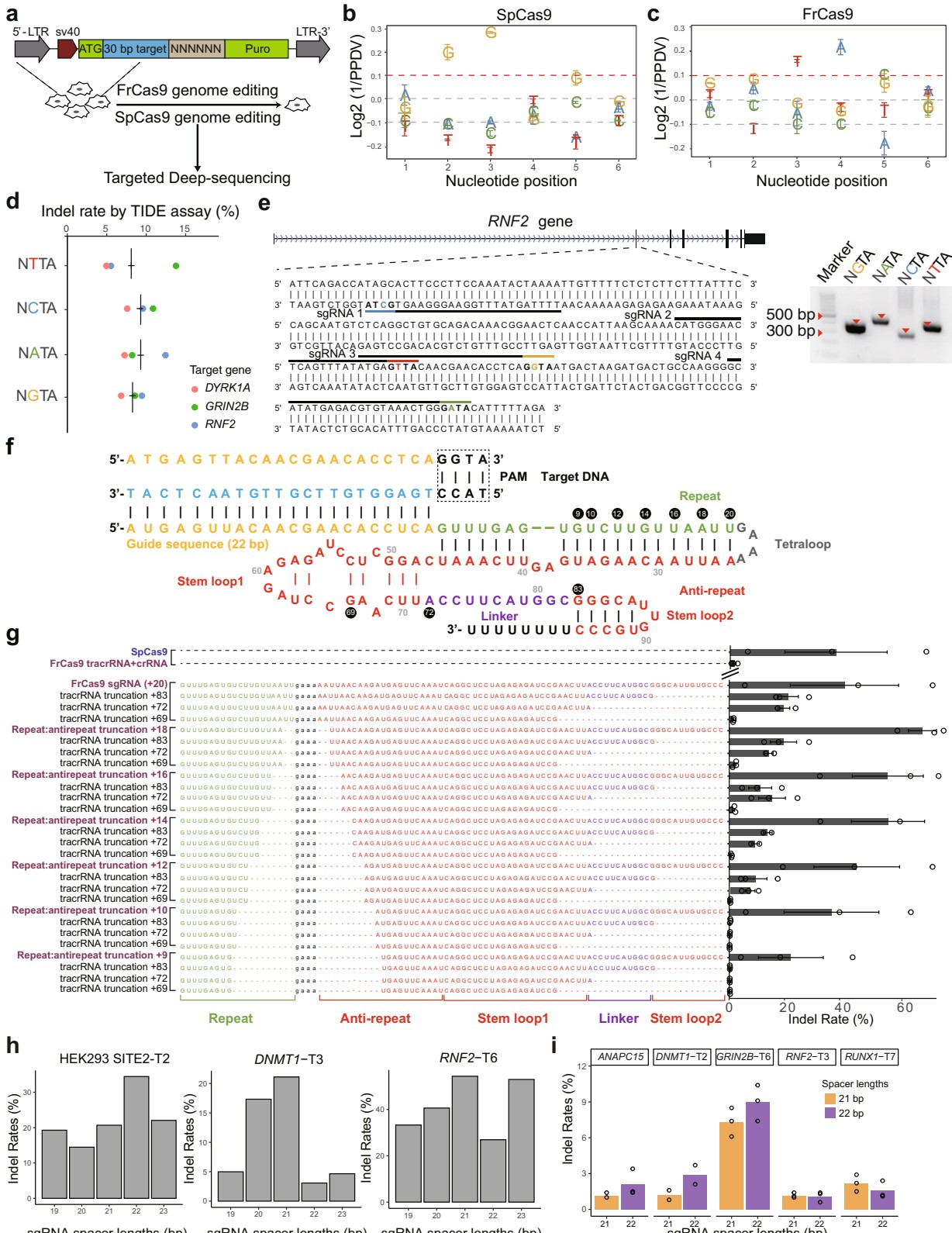

tracrRNA tails decreased the efficiency of FrCas9 dramatically (Fig. 2g). Meanwhile, the truncation of 3′ crRNA affected the cleavage ability modestly when truncated from +18 to +12 (Fig. 2g). The gene editing by FrCas9 at other two endogenous sites (*GRIN2B*-T3 and *GRIN2B*-T9) confirmed above conclusions (Supplementary Fig. 5). Notably, in *RNF2* site, the editing

efficiency of FrCas9 was higher than that of SpCas9 (Fig. 2g, 70.08% vs. 38.36%).

Next, to test the optimal length of FrCas9 guide RNAs, we designed the 19–23 bp guide sequences at three sites (Fig. 2h and Supplementary Table 5). All 5 lengths effectively edited the targets, but the highest genome editing efficiencies were achieved in 21-22 bp (Fig. 2h, *DNMT1-T3* 21 bp = 21.0%, *RNF2-T6* 21

**Fig. 2 FrCas9 is active in eukaryotic cells. a** The schematic of puromycin depletion assay. **b, c** The PAM results of SpCas9 and FrCas9 from the puromycin depletion assays. Data are presented as mean values ± S.E.M. ($n = 3$ biological independent replicates). **d** The TIDE assay showed FrCas9-induced indel rates in 3 human genes by 12 sgRNAs, which differed in the 2nd PAM base. **e** Genome editing by FrCas9 in the human *RNF2* gene, validated by double-stranded oligodeoxynucleotide (dsODN) breakpoint PCR. And the Sanger sequencing was in Supplementary Fig. 2. Uncropped gel image is provided in Source Data. **f** The schematic representation of the sgRNA:target DNA complex. **g** The efficiency of sgRNAs with truncated scaffolds of FrCas9 assayed by target amplicon sequencing. Data are presented as mean values ± S.E.M. ($n = 3$ biological independent replicates). **h** The FrCas9 editing efficiencies with 19–23 bp spacer lengths in three human sites by target amplicon sequencing. **i** The FrCas9 editing efficiencies with 21 and 22 bp spacer lengths in five human sites by TIDE ($n = 3$ biological independent replicates). Source data are provided with this paper.

bp = 54.39%, HEK293 SITE-T2 22 bp = 34.45%). To compare the editing efficiencies of 21 bp and 22 bp, we involved additional 5 targets and observed that 22 bp showed better editing efficiency than 21 bp at 3 sites (Fig. 2i, *GRIN2B*-T6, 8.95% vs. 7.3%; *DNMT1*-T2, 2.9% vs. 1.2%; *ANAPC15*, 2.1% vs. 1.2%). Therefore, the optimal length of FrCas9 guide RNA was 22 bp, which was in consistent with the small RNA-seq results (Fig. 1d).

The tolerance of mutations in guide sequences was related to the specificity of Cas9 orthologs. Next, we investigated the sgRNA specificity of FrCas9 using 22 sgRNAs, all of which contained a single nucleotide mutation (Supplementary Fig. 6 and Supplementary Table 5). The results showed that mutated sgRNAs had no obvious cleavage, suggesting that FrCas9 has long seed region and possesses high specificity (Supplementary Fig. 6).

**FrCas9 genome editing shows high specificity and activity.** Since SpCas9 and FrCas9 both have 2-nucleotide PAMs (SpCas9: 5′-NGG-3′, FrCas9: 5′-NNTA-3′), we compared their genome editing efficiency and specificity in sequence with 5′-GGTA-3′. Based on this principle, we selected 11 human endogenous sites with 5′-GGTA-3′ PAM (Supplementary Table 6) and compared the cutting efficiency and off-target effect of SpCas9 and FrCas9 using GUIDE-seq[4].

First, we detected dsODN integration in all 11 sites by dsODN-breakpoint PCR (Supplementary Fig. 7). The GUIDE-seq experiments confirmed the most frequent locations of dsODN incorporation of FrCas9 and SpCas9 were both the 3rd or 4th base upstream of PAM (Fig. 3a, FrCas9 3rd: $n = 5$, 4th: $n = 5$; SpCas9, 3rd: $n = 3$, 4th: $n = 4$). Totally, among the 11 sites, only one FrCas9 off-target was detected in *GRIN2B*-T3 site, while 2-3 SpCas9 off-targets per sgRNA were detected in HEK293 SITE-T2, *DYRK1A*-T2, *GRIN2B*-T8, *GRIN2B*-T9 sites (Fig. 3d). Importantly, we observed a significant higher on:off ratio (defined as on:off target reads) of FrCas9 than SpCas9 at 11 sites (Fig. 3e, $P < 0.05$, paired *Wilcox* test). The above results indicated that FrCas9 genome editing showed high specificity and activity. The same trend was also observed in U2OS cell line (Supplementary Fig. 8).

Further, we compared the efficiency and specificity of FrCas9 with SpCas9 and its high-fidelity version (SpCas9-HF1, HiFi-Cas9 and eSpCas9) on *DYRK1A*-T2 and *GRIN2B*-T9 sites. In *DYRK1A*-T2 site, the off-targets of each variant were FrCas9 (0), SpCas9 (15), SpCas9-HF1 (2), HiFi-Cas9 (2) and eSpCas9 (1), respectively. The off-targets in *GRIN2B*-T9 site were as below: FrCas9 (0), SpCas9 (8), SpCas9-HF1 (6), HiFi-Cas9 (12) and eSpCas9 (7) (Fig. 3f, g). As expected, the FrCas9 exhibited the highest on:off ratio in both sites (Fig. 3h).

**FrCas9 can be used to target HPV genomes.** We next set out to test the efficiency and specificity of FrCas9 as a potential in targeting HPV genomes[27]. For HPV 18 genome, we observed a nearly 100% target coverage of FrCas9 with sgRNA distributed per of 5.65 bp in average (Fig. 4a, b), which was greater than

SpCas9 (81.17% coverage and 8.32 bp mean distances of sgRNA distribution).

To evaluate the efficacy of SpCas9 and FrCas9 in HPV 18, we conducted GUIDE-seq in 19 HPV sites and each site contained overlapping sgRNAs for SpCas9 and FrCas9 (Fig. 4d, e). All 19 sites showed editing activities with both SpCas9 and FrCas9 and the efficiencies were comparable between two Cas9 (Fig. 4f). SpCas9 sgRNAs had average 35.58 off-targets per sgRNA while FrCas9 sgRNAs had average 1.68 off-targets per sgRNA (Fig. 4g). Based on the on/off-target ratios represented by the GUIDE-seq reads, FrCas9 exhibited high efficiency and specificity in HPV 18 gene editing (Fig. 4h, $P < 0.0001$, two-sided paired *Wilcox* test).

Further, we selected the sgRNAs targeting HPV URR and E7 to investigate the apoptosis induced by FrCas9 in HPV 18 positive HeLa cell line. Compared to the 13.74% apoptosis rate of the negative controls, FrCas9 with the URR and E7 sgRNAs achieved 26.89% and 36.7% apoptosis rates, respectively (Fig. 4i). Therefore, FrCas9 can be used in targeting HPV 18.

**FrCas9 has characteristics for wide applications in genome engineering.** Since the 5′-NNTA-3′ PAM of FrCas9 has distinct targets for correcting human pathogenic variants, we repurposed FrCas9 for application of base editing. First, three point-mutations of E796A, H1010A and D1013A were respectively incorporated to generate different FrCas9 nickases (nFrCas9)[28] (Supplementary Fig. 9a). Then, we combined E796A nFrCas9 with the optimized fourth-generation cytidine base editor BE4Gam[29] and seventh-generation adenine base editor ABE7.10[30]. We observed that the editing window of FrCas9-BE4Gam was 6th–10th bases and that of FrCas9-ABE7.10 was 6th–8th bases (Supplementary Fig. 9b, c). Based on the above characteristics, we calculated the targeting scopes of FrCas9-BE4Gam and FrCas9-ABE7.10 in ClinVar databases[31]. For pathogenic mutations that could be precisely corrected by FrCas9-BE4Gam, 90.38% (235/260) events were different from SpCas9-BE4Gam. For pathogenic mutations that could be precisely corrected by FrCas9-ABE7.10, 92.21% (1196/1297) events were different from SpCas9-ABE7.10 (Fig. 5a). Therefore, the TA-rich PAM greatly expanded the targets in human genome for base-editor to correct human disease-associated mutations.

Notably, the PAM of FrCas9 (5′-NNTA-3′) is palindromic, which offers pairwise "back-to-back" existence of sgRNAs (Fig. 5b). This feature could broaden the scopes of FrCas9 base-editors by modifying two close editing windows at the same time (Fig. 5b) and increase the target distribution and density of FrCas9 sgRNAs. We calculated the 5′-GG-3′ (represented for SpCas9 PAM) and 5′-TA-3′ (represented for FrCas9 PAM) distributions in human genomes (Fig. 5c, d). Compared to SpCas9 (median = 5 bp, mean = 8.66 bp)[5], FrCas9 showed more intensive distributions (median = 1 bp, mean = 6.16 bp) in human genomes, providing additional applicable loci.

Interestingly, the TA palindromic PAM also has multiple sites on TATA box (Fig. 5e), which is a crucial promoter element for eukaryotic organisms[32,33]. We tested FrCas9 CRISPR interference (CRISPRi) in three TATA-box promoted genes, *ABCA1*, *UCP3*

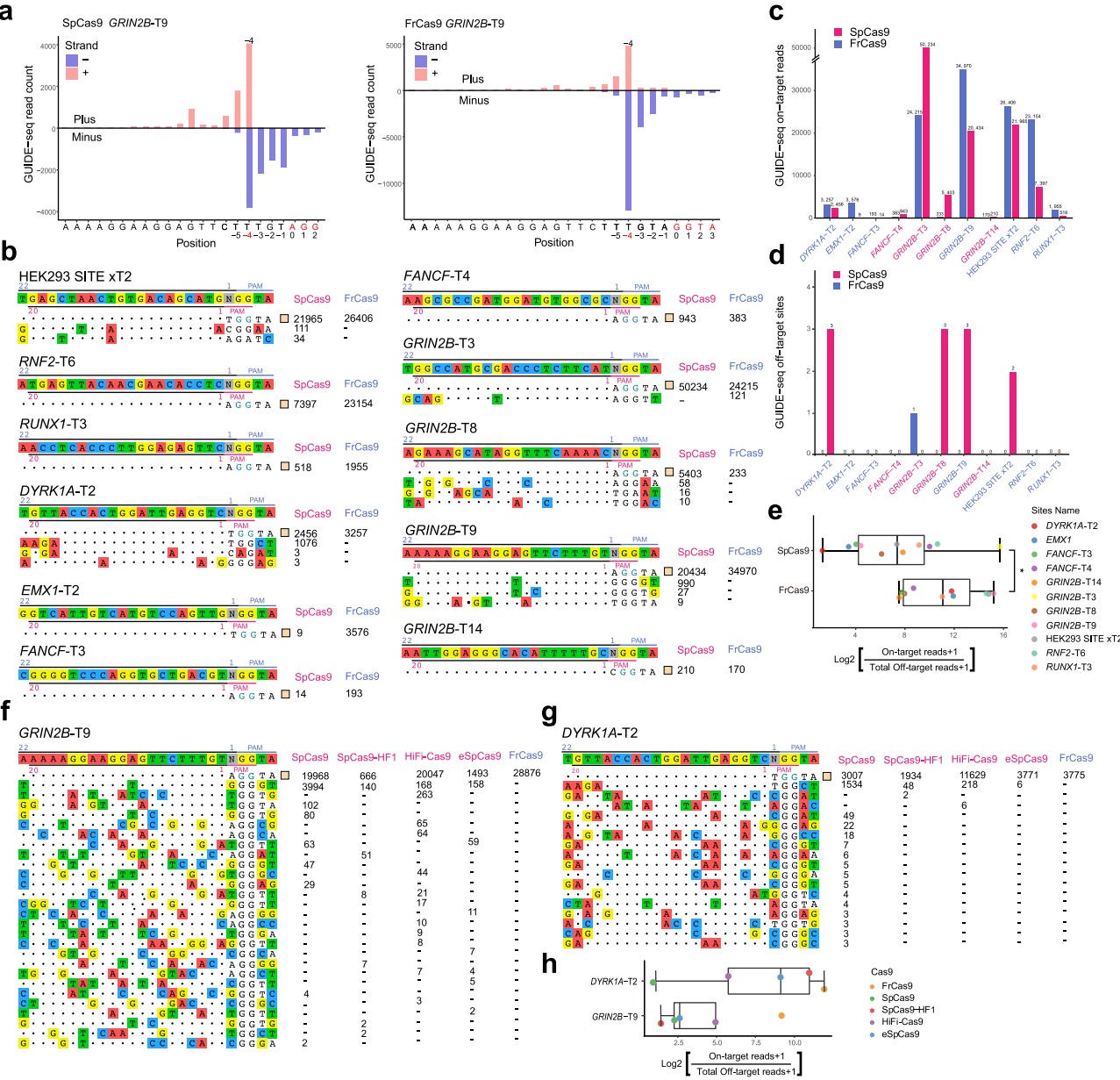

**Fig. 3 The genome-wide specificities of FrCas9 and SpCas9. a** The start mapping positions of GUIDE-seq reads for SpCas9 and FrCas9 targeting *GRIN2B*-T9. The 1st bases of PAM sequences (red) were position 0 and the most frequent dsODN incorporation sites were colored in red. **b** The off-targets of SpCas9 and FrCas9 for 11 sites, generated by GUIDE-seq in HEK293T cells. The sgRNA and PAM ranges of SpCas9 (20 nt sgRNA and 3 nt PAM) and FrCas9 (22 nt sgRNA and 4 nt PAM) were marked. GUIDE-seq read counts of each site were shown on the right side. **c** Summary of GUIDE-seq on-target reads of SpCas9 and FrCas9 at the above 11 sites. **d** Summary of off-target counts of SpCas9 and FrCas9 at the above 11 sites. **e** The on:off ratio of GUIDE-seq reads. *N* = 11 sites. Box plots indicate median (middle line), 25th, 75th percentile (box) and 5th and 95th percentile (whiskers). *P* = 0.032, two-sided Student's *t*-test, * representing *P* < 0.05. **f**, **g** The off-targets of FrCas9, SpCas9, SpCas9-HF1, HiFi-Cas9 and eSpCas9 in *DYRK1A*-T2 (**f**) and *GRIN2B*-T9 (**g**) detected by GUIDE-seq in HEK293T cell line. **h** The on:off ratio of FrCas9, SpCas9, SpCas9-HF1, HiFi-Cas9 and eSpCas9 in *DYRK1A*-T2 and *GRIN2B*-T9. *N* = 5 Cas9 nucleases. Box plots indicate median (middle line), 25th, 75th percentile (box) and 5th and 95th percentile (whiskers). The ratio is defined by GUIDE-seq on-target reads dividing total off-target reads. Source data are provided with this paper.

and *RANKL* (Fig. 5e). By cleaving the TATA-box, FrCas9 reduced *ABCA1*, *UCP3* and *RANKL* expression by 31.37%, 49.91% and 39.62%, respectively. Meanwhile, the expression of *ABCA1*, *UCP3* and *RANKL* decreased 61.67%, 45.61% and 42.60% by dFrCas9 directly binding to the TATA-box, respectively (Fig. 5f). Together, FrCas9 possesses potential for efficient genome engineering of TATA-box related genetic diseases.

Further, we tested FrCas9 CRISPR activation (CRISPRa) using dFrCas9-VP64 directly targeting the TATA-box, and compared its performance with dSpCas9-VP64 targeting the upstream of TATA-box. The CRISPRa experiments were conducted in *ABCA1*, *SOD1*, *GH1* and *MBL2* genes. The results showed that dFrCas9-VP64 enables effective transcriptional activation. Moreover, the fold activation of dFrCas9-VP64 in *ABCA1*, *GH1* and *MBL2* was higher than that of dSpCas9-VP64, while the fold activation of *SOD1* gene was comparable to that of dSpCas9-VP64 (Fig. 5g). Therefore, FrCas9 is a promising tool for CRISPR screening due to its 5'-NNTA-3' PAM.

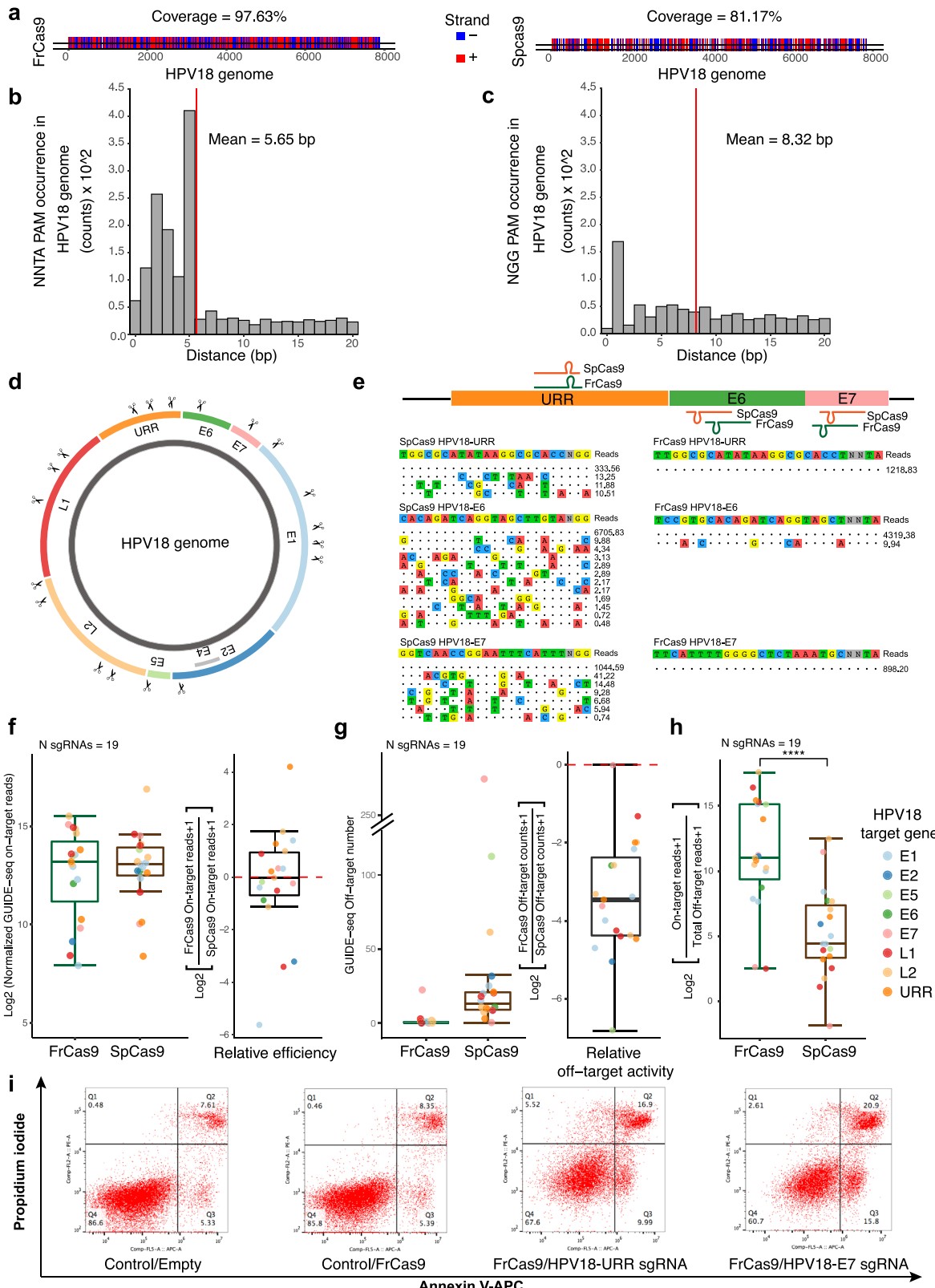

## Discussion

The editing efficiency and specificity are two important factors for the applications of genome engineering tools. In this study, the FrCas9 from *Faecalibaculum rodentium* Type II-A CRISPR/Cas systems was fully characterized and validated. FrCas9 hold high efficiency and specificity with simple 2-nuleotide PAM sequences

(5′-NNTA-3′). Therefore, FrCas9 will be a powerful and safe platform for biotechnological and clinical research.

Interestingly, the PAM of FrCas9 (5′-NNTA-3′) was palindromic, which resulted in the special advantage that its targets occurred in pairwise with two opposite orientations. This characteristic expands FrCas9 targeting scope in genomes, decreases

**Fig. 4 FrCas9 is promising in anti-HPV18 treatments. a** The target coverages of FrCas9 and SpCas9 in HPV18 genomes. **b** The sgRNA distribution densities for FrCas9 in HPV18 genomes. **c** The sgRNA distribution densities for SpCas9 in HPV18 genomes. **d** The 19 sites distribution in HPV18 genome for GUIDE-seq comparisons between FrCas9 and SpCas9. **e** The schematic illustration and GUIDE-seq results of SpCas9 and FrCas9 targeting URR, E6 and E7 genes. The reads number was normalized. **f** The GUIDE-seq normalized on-target reads of FrCas9 and SpCas9 in 19 sites. **g** The GUIDE-seq off-target counts of FrCas9 and SpCas9 in 19 sites. **h** The on-target vs. off-target ratios of FrCas9 and SpCas9 (N =19 sites, $P = 0.0000267$, two-sided paired *Wilcox* test, **** representing $P < 0.0001$). Box plots indicate median (middle line), 25th, 75th percentile (box) and 5th and 95th percentile (whiskers). The ratio is defined by GUIDE-seq on-target reads dividing total off-target reads. **i** The FrCas9 induced cell apoptosis in HPV18 positive HeLa cell line. Source data are provided with this paper.

the distances between average two PAMs, and increases the editing coverage window (Figs. 4a–c and 5b, c), enabling greater potentials in genome therapeutics. For instance, the target coverages of FrCas9 in HPV 16 and HPV 18 achieved 98.92% and 97.63%, respectively, which were wider than SpCas9 (HPV 16 = 70.35%, HPV 18 = 81.17%). For precise modification including base editing and prime editing, the palindromic PAM may provide more targeting flexibility and editing range, both are crucial factors for therapeutic efficacy.

It is worth noting that the 5′-NNTA-3′ PAM has optimal targets in TATA box (core sequence: 5′-TATAAATAAT-3′), which is a conserved promoter element for eukaryotic organisms and is related to various kinds of human hereditary diseases[32,33]. On the one hand, this TA-containing PAM could help FrCas9 block the abnormal upregulation of pathogenic genes by decreasing or destroying the TATA box-induced transcription. On the other hand, if mutated TATA box downregulated the expressions of normal functional genes, FrCas9 may repair the TATA box by homologous recombination (HDR) or prime editing to correct the genetic disorders. Future investigations are warranted in this direction.

Taken together, FrCas9 with 5′-NNTA-3′ PAM sequence has high efficiency and specificity. We envision that this CRISPR system will not only benefit the biological and clinical research in this field, but will also provide deeper understanding toward the mechanism of CRISPR-mediated genome engineering.

## Methods

**Bioinformatical predictions of FrCas9 functions and features**. FrCas9 was first detected and deposited in the CasPDB database[34] but full characterization of this Type II-A CRISPR/Cas system has not been studied. First, we downloaded the genome of *Faecalibaculum rodentium* strain NYU-BL-K8 (Accession code: NZ_MPJZ01000004.1). Then, we annotated the genome with standalone CRISPRCasFinder software (version: 2.0.2)[35]. After locating the *cas* genes and CRISPR arrays, we built python scripts to search the potential tracrRNA expression regions based on reported protocol[21]. By predicting the secondary structures of crRNA and tracrRNA with NUPACK software[36] (http://www.nupack.org), we confirmed the tracrRNA located upstream *cas9* gene. For FrCas9 protein, we used the HHpred software (https://toolkit.tuebingen.mpg.de/tools/hhpred)[37] to predict the functional domains, and based on the prediction results we extracted FrCas9 PAM Interaction (PI) domain. The phylogenetic trees of Cas9 orthologs and PI domains were constructed using FastTree software (version: 2.1.11)[38] with the "-wag -gamma" parameter after alignment by MAFFT software (version: 7.490)[39]. The PAM sequences of each Cas9 ortholog in phylogenetic trees were made by Weblogo3 (http://weblogo.threeplusone.com/) using matrixs[40–42].

**Bacterial RNA sequencing and analysis**. The *E. coli* carrying the pET-28A-FrCas9 locus was grown 16 h at 37 °C in LB (Luria Bertani) medium supplemented with 50 μg/ml kanamycin. Total RNA was purified using Direct-Zol RNA kit, followed by DNase I treatment, 3′ dephosphorylated by T4 PNK (M0201L) and ribosomal removal by Ribo-Zero rRNA Removal Kit. Finally, the small RNA Library Prep Set was used for sample preparation. The library was then sequenced using illumina platform with pair-end 150 mode. The RNA sequencing data was then aligned to the reference of pET-28A-FrCas9 plasmid using STAR software[43]. The stranded bedGraph format was then transformed from alignment results to generate the visualization of tracrRNA and crRNA expression by Sushi R package (version: 3.14)[44].

**Cell culture**. HeLa (CCL2, ATCC), U2OS (HTB-96, ATCC) and HEK293T (CRL3216, ATCC) cell lines were purchased from ATCC (American Type Culture

Collection, VA, USA). Short tandem repeat (STR) testing was performed to confirm cell identity. These cells were cultured in Dulbecco's Modified Eagle Medium (C11995500BT, GIBCO) supplemented with 10% fetal bovine serum (10270-106, GIBCO) and incubated at 37 °C and 5% $CO_2$ in a constant temperature incubator. Cell passaging was performed at a 1:3 split ratio when the cells reached 90% confluence.

**DNA extraction**. Genomic DNA in this study was extracted with a Kit (EE101-11, TransGen, China).

**Plasmid depletion assay and analysis**. To generate the plasmid library, single-stranded DNA oligonucleotides containing a protospacer flanked with 6-nucleotide (6 N) randomized PAM sequences were synthesized (Genewiz, China) and extended with Klenow Fragment (3′→5′ exo-) (M0212S, NEB). Then, the plasmid library ($n = 4096$) was assembled to pACYC184 backbone (AmpR+) by Gibson assembly[45]. After assembly, the plasmid library was transformed into *E. coli* DH5α followed by the extraction of plasmids. We transfected 200 ng above plasmid library into electrocompetent *E. coli* which harbored a pET-28A-FrCas9-array locus or a pET-28A control plasmid (Kana+). Then, the transfected *E. coli* was selected with dual-antibiotics (50 μg/ml kanamycin and 100 μg/ml ampicillin) on LB medium for 30 h at 25 °C. Finally, plasmid DNA was extracted, and the PAM region was PCR amplified and sequenced using illumina platform in pair-end 150 mode. The targets and primers used were listed in Supplementary Tables 5 and 7.

For analysis, we extracted 6 N PAM sequences with in-house scripts and for each PAM sequence we calculated the Post-selection PAM Depletion Value (PPDV), which was the ratio of the post-selection frequency of a PAM in the pET-28A-FrCas9 population divided by the frequency of that PAM in control library[46]. For PAM weblogo visualization, Weblogo3 was used to generate logo based on top 10% depleted PAM sequences ranked by the depletion efficacy (reversely correlated with PPDVs). For PAM wheel construction, the positive depleted PAMs were used to calculate the relative depletion of three nucleotides (2-4 PAM positions) as a previous study reported[14].

**Plasmid interference assay**. The same protospacer used in plasmid depletion assay flanked by the identified "NRTA (GGTA)" PAM was synthesized (Genewiz, China) and cloned into pUC19 backbone (AmpR+) by Gibson assembly. Then, 10 ng plasmid was transfected into the electrocompetent *E. coli* which carried the pACYC184-FrCas9 locus or pACYC184 control plasmid. A series dilution was plated on dual-antibiotics selective medium (25 μg/ml chloramphenicol and 100 μg/ml ampicillin) at 37 °C for 16 h. The growth of colony units was then observed between groups. The targets primers used for Gibson assembly were listed in Supplementary Tables 5 and 7.

**Plasmid constructions for eukaryotic validation**. The SpCas9 plasmid used in this study was purchased from Addgene (Plasmid #42229). Then, the SpCas9 sequence was amplified and cloned into PX330 vector. The sgRNA sequences were synthesized by GENEWIZ Biotech Co. (Suzhou, China) and cloned into PXZ vector.

**Puromycin depletion assay in vivo and analysis**. A lentiviral plasmid containing SV40 promoter and expressing puromycin-N-acetyltransferase (PAC) was used to generate the pooled library for puromycin depletion assay. Specifically, the 66 bp sequences (consisting of 30 bp protospacer, 6N randomized PAM sequences and 30 bp homology arms sequences) were inserted after the initial codon (ATG) by Gibson assembly, which caused the in-frame mutation and did not influence the function of PAC puromycin resistant gene[47]. Lentiviral products were obtained by co-transfection of library plasmids with three viral packaging plasmids into HEK293T using the polyethylenimine (PEI) method. HEK293T cells were grown in 10 cm dishes to 80–90% confluency. For each dish, transfection was performed using 30 μl of PEI (Promega), 4.2 μg of pLP1, 2.1 μg of pLP2, 3.1 μg of pVSVG, and 4 μg of library plasmids. After 30 min of incubation at room temperature, the mixture was added to the HEK293T cells. After 72 h of infection, the virus supernatant was harvested and passed through a 0.45 μm filter. Then, the HEK293T cell line was seeded at 400,000 cells in six-well plates and the next day transduced with a mixture of DMEM and virus supernatant at a ratio of 1:1. After

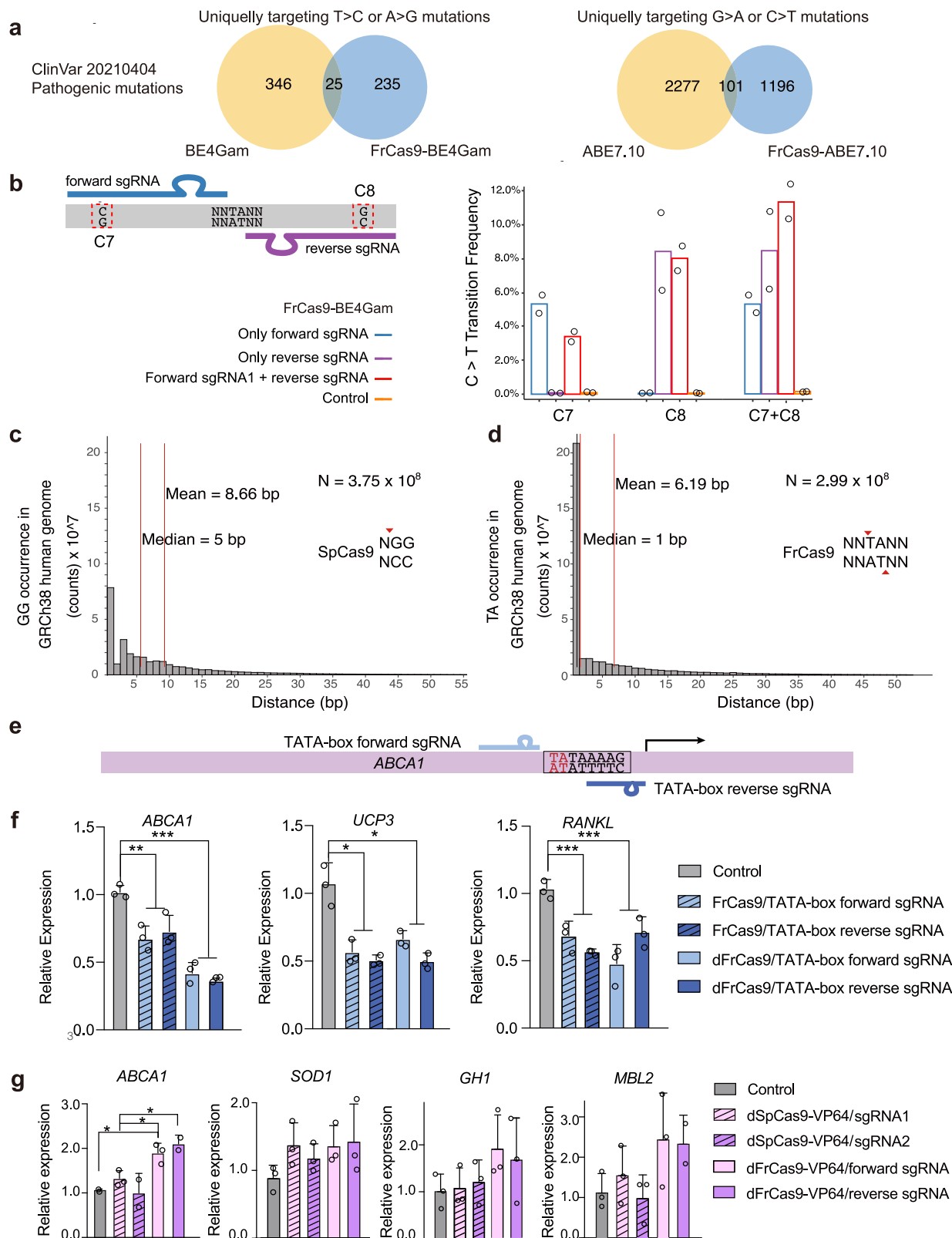

12–16 h of incubation, the cells were selected in the presence of puromycin (1 μg/ml) for 3 days.

The transduced cells were split into three groups according to the transfected plasmids: control (2.5 μg empty PX330 vector), SpCas9 (2.5 μg PX330-SpCas9-sgRNA plasmid), and FrCas9 (2.5 μg PX330-FrCas9-sgRNA plasmid), respectively, each with three replicates. To avoid the correct repair after the PAC gene was edited, we also co-transfected 1.5 μl dsODN (carried with stop codon) by using SF Cell Line 4D-Nucleofector™ X Kit (V4XC-2024, Lonza, Germany) in above

experiment groups. After puromycin selection (1.5 μg/ml) for 5 days, cells were collected separately for genomic DNA extraction, followed by PCR amplification of the protospacer-PAM region and deep-sequencing. The targets and primers used were listed in Supplementary Tables 5 and 7. The PPDV calculation method from deep-sequencing data was the same as that of plasmid depletion assay.

**HT-PAMDA experiments and data analysis.** We purchased the p11-Cas9_random_PAM-site1 plasmid (Addgene #160132) and pCMV-T7-SpCas9-

**Fig. 5 The wide applications of FrCas9 due to its 5′-NNTA-3′ PAM. a** The veen diagram of pathogenic mutations in ClinVar database that could be corrected by SpCas9 and FrCas9 base editors. **b** The C > T transition efficiencies of FrCas9-BE4Gam using 2 "back-to-back" sgRNA at the same time. The efficiencies were assayed by amplicon sequencing (n = 2 biological independent replicates). **c** The 5′-GG-3′ (represented for SpCas9 PAM) and **d** The 5′-TA-3′ (represented for FrCas9 PAM) distribution in the GRCh38 human genome. **e** The schematics of FrCas9 targeting TATA-boxes of *ABCA1* gene. CRISPRi **f** and CRISPRa **g** of FrCas9 and SpCas9 by targeting TATA-boxes. The experiments were conducted in HEK293T cells and expression was quantified by qPCR. Data are provided as mean value ± S.D (n = 3 biological independent replicates). The *p* values of Control vs. FrCas9/TATA-box sgRNAs on *ABCA1*, *UCP3*, and *RANKL* were 0.0010, 0.017, 0.00038, respectively. The *p* values of Control vs. dFrCas9/TATA-box sgRNAs on *ABCA1*, *UCP3*, and *RANKL* were 0.00024, 0.015, 0.00098, respectively. The *p* values of Control vs. dFrCas9-VP64 forward sgRNA, dSpCas9-VP64 sgRNA1 vs. dFrCas9-VP64 forward sgRNA and dSpCas9-VP64 sgRNA1 vs. dFrCas9-VP64 reverse sgRNA on *ABCA1* were 0.021, 0.026, 0.042, respectively (*P < 0.05, **P < 0.01, ***P < 0.001, two-sided Student's *t*-test). Source data are provided with this paper.

P2A-EGFP plasmid (Addgene #139987). Further, we constructed the pCMV-T7-FrCas9-P2A-EGFP plasmid and synthesized the in vitro transcription templates of SpCas9 and FrCas9 sgRNAs for PAM-site1. Then, we conducted the HT-PAMDA experiments with timepoints including 1 min, 8 min and 32 min. The PAMDA procedures were performed according to the protocol[24]. We generated high-throughput data with about sequencing depth of ~15,000,000 reads per timepoint, assuring sufficient coverage to resolve up to 5 nt of PAM preference. The data analysis was used the scripts deposited in Github (https://github.com/kleinstiverlab/HT-PAMDA).

**TIDE assay and analysis**. TIDE analysis was performed using available software (tide.nki.nl). The targets and primers used for PCR reactions were listed in Supplemental Tables 5 and 7.

**dsODN breakpoint-PCR**. The PCR reaction was conducted with one primer upstream/downstream target sites and the other primer attached to dsODN sequences (Green Taq Mix, P131-01, Vazyme). The targets and primers used were listed in Supplementary Tables 5 and 7.

**Targeted amplicon sequencing and Indel frequency analysis**. We used illumina Hiseq2500 platform to generate paired-end sequencing data with read length of 150 bp. For data analysis, after quality control of the raw sequencing data by fastp with default parameters[48], pair-end reads were merged into one read by FLASH software[49]. Then, all merged reads were aligned to the references consisted of product sequences with BWA-MEM[50]. The rewritten python script from CRISPRMatch package[51] was used to extract sequences of the targets ±10 bp flanking regions and calculate the indel (insertion and deletions) rates. Control experiments were conducted without adding nucleases to exclude the indels caused by background mutation or sequence errors. The cutting efficiency of each site was represented by mean indel rate of experiment group subtracting that of control group. The targets and primers used were listed in Supplementary Tables 5 and 7.

**Lentiviral library generation of point mutation sgRNAs**. To test the specificity of FrCas9, we generated a lentiviral library for sgRNAs containing single-point mutated guide sequences. A plasmid of hU6 promoter expressing puromycin resistant gene was used to construct a sgRNA-target plasmid library by Gibson assembly. Then, the lentiviral transduced HEK293T cells were obtained and maintained by the same approach in puromycin depletion assy.

For comparison, transduced cells were split into two groups according to transfection plasmids: control (2.5 μg empty PX330 vector) and FrCas9 (2.5 μg PX330-FrCas9 plasmid), each with three replicates. Transfection was performed using SF Cell Line 4D-Nucleofector™ X Kit (V4XC-2024, Lonza, Germany). After 72 h, cells were collected separately for genomic DNA extraction, followed by PCR amplification of the sgRNA-target region and deep-sequencing. The targets and primers used were listed in Supplementary Tables 5 and 7.

For analysis, the indel rate of each single-point mutated sgRNA was calculated and corrected by multiplying the sgRNA evenness index derived from amplification sequencing data of sgRNA region in the transduced HEK293T cells:

$$\text{sgRNA evenness index} = \frac{Total\ read}{N\ x\ sgRNA\ Reads} (\text{N represented for sgRNA counts})$$

**GUIDE-seq experiments and analysis**. The same double-stranded oligodeoxy-nucleotide (dsODN) was synthesized by Sangon Biotech Co. (Shanghai, China). First, we co-transfected 5 μl dsODN and 10 μg PX330-SpCas9-sgRNA/PX330-FrCas9-sgRNA plasmids into HEK293T cell line using SF Cell Line 4D-Nucleo-fector™ X Kit (V4XC-2024, Lonza, Germany) or into U2OS cell line using SE Cell Line 4D-Nucleofector™ X Kit (V4XC-1024, Lonza, Germany).

After 72 h, cells were harvested for DNA extraction, followed by dsODN-PCR verification of effective cleavage. Then, GUIDE-seq libraries were constructed as previous study reported[52]. Briefly, the DNA went through shearing, adding Y adapters and two round of PCR, and were finally sequenced using MGISEQ-2000RS sequencer with customized settings for 16 bp UMI. Data was first

demultiplexed using in-house python scripts and then analyzed using guideseq v1.1[53]. The off-targets were identified using the original standards with mismatches ≤ 7[4]. The targets and primers used were listed in the Supplementary Tables 5 and 7.

**Apoptosis analysis**. HeLa cells were seeded into 12-well plates and transfected with 2.5 μg FrCas9-gRNA plasmid using SE Cell Line 4D-Nucleofector™ X Kit. At 48 h post transfection, cells were collected and stained using an Annexin V-APC/PI Apoptosis detection kit (KGA1030, Key GEN BioTECH, China).

**Real-time PCR (qPCR)**. The PCR was conducted using the Bio-Rad CFX96 system with PowerUp™ SYBR™ Green Master Mix (A25742, Applied Biosystems, USA). The relative expression change was calculated using the comparative Ct method, which compared the Ct value of the no sgRNA group and the TATA-box targeting sgRNA group. The primers used were listed in Supplementary Table 7.

**Statistical analysis**. Comparison between groups were using R Programming Language 3.5.2. P < 0.05 was statistically significant (*P < 0.05, **P < 0.01, ***P < 0.001, ****P < 0.0001).

**Reporting summary**. Further information on research design is available in the Nature Research Reporting Summary linked to this article.

## Data availability

Source data are provided with this paper. The genome of *Faecalibaculum rodentium* strain NYU-BL-K8 is available in NCBI database under accession code NZ_MPJZ01000004.1. The sequencing data generated in this study has been deposited in the NCBI Sequence Read Archive (SRA) database under the accession code PRJNA766437. Source data are provided with this paper.

## Code availability

The FrCas9 identification pipeline was deposited in Github (https://github.com/Freya-Cui-2020/FrCas9).

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

## Acknowledgements

This work was supported by the National Natural Science Foundation of China (grant no. 32171465, Z.Hu. and 82102392, R.T.); The National Science and Technology Major Project of the Ministry of science and technology of China (grant no. 2018ZX10301402, Z.Hu.); General Program of Natural Science Foundation of Guangdong Province of China (grant no. 2021A1515012438, Z.Hu.); the National Postdoctoral Program for Innovative Talent (grant no. BX20200398, R.T.); the China Postdoctoral Science Foundation (grant no. 2020M672995, R.T.); Guangdong Basic and Applied Basic Research Foundation (grant no. 2020A1515110170, R.T.); Characteristic Innovation Research Project of University Teachers (grant no. 2020SWYY07, R.T.); the National Ten Thousand Plan-Young Top Talents of China (Z.Hu.).

## Author contributions

Z.Hu., Z.C., R.T., Z.Huang., Z.J. and H.X. identified CRISPR loci using bioinformatic pipeline. H.X., D.L., R.Z. and H.M. performed the experiments. Z. Hu., Z.C. and H.X. drafted the manuscript. L.L., J.L., Zhe.H., B.L., B.M. and H.W. revised the manuscript.

## Competing interests

A related patent application on the pipeline of identifying potential CRISPRs has been filed (patent applicant: Sun Yat-sen University; the First Affiliated Hospital of Sun Yat-sen University, inventors: Z.Hu., Z.C., R.T., Z.J., Z.Huang. and M.L., application number: 202110589533.8). The remaining authors declare no competing interests.
