## [Peer Review File · Nature Communications]

Reviewers' Comments:

Reviewer #1:

Remarks to the Author:

In this work, Cui, et al. identify a novel Cas9 from *Faecalibaculum rodentium*, which simultaneously possesses broad 5'-NNTA-3' PAM specificity, efficient editing in human cells, and high specificity in relation to off-targets. They characterize their enzyme using bioinformatic analysis on the *Faecalibaculum rodentium* genome, PAM depletion assays in bacterial cells for PAM determination, GUIDE-Seq for off-target analysis, and further mismatch tolerance lentiviral assays for specificity analysis. Furthermore, they engineer base editors incorporating FrCas9 and show high base editing efficiency. More interestingly, FrCas9 has a palindromic PAM, enabling targeting of pairwise targets in opposite orientations, opening up a more significant percentage of genomes as compared to SpCas9. Finally, the authors demonstrate effective editing at TATA boxes, which are highly amenable to FrCas9 editing.

Overall, while this work is not very novel, as there are Cas9s with minimal PAMs (SpRY, iSpyMac, Sc++, ScCas9), some of whom can edit overlapping targets with similar efficiency and accuracy, augmenting the list of viable CRISPR tools is always needed. Importantly, this work covers most of the critical assays for Cas9 characterization and is in overall good shape for eventual publication in *Nature Communications*. However, I would like the authors to address my concerns by conducting a few more validation assays, as detailed below.

Major Concerns

1. The authors utilize a depletion assay to assess PAM specificity. However, from the text and figures, it is not clear about the details of the assay and whether it would faithfully recapitulate the PAM sequences of other known Cas9 effectors. In addition to providing more detailed text within the manuscript, I would ask the authors to conduct an additional PAM determination assay (PAMDA) that has recently been published and evaluate PAM sequences in a more relevant context: <https://www.nature.com/articles/s41596-020-00465-2>. I also ask that they conduct comparisons with a few other Cas9s of their choosing for validation.
2. To confirm that editing rates on 5'-NNTA-3' PAM sequences in human cells are robust, I would suggest the authors do characterization on at least 24 targets in 8 different gene loci. Most papers do 48+ targets to show relevant editing, but as the authors demonstrate editing in other contexts, 24 different targets should suffice.
3. The authors use GUIDE-Seq to evaluate on- and off-target efficiency. GUIDE-Seq is an internally controlled assay, not one that can be compared across samples. Thus, I would advise the authors to not use GUIDE-Seq to determine editing rate on target, but rather to calculate the ratio of on:off target reads as a measure of specificity at each site. Plotting these values across the different sites will provide a fuller understanding of FrCas9's specificity vs. SpCas9. Furthermore, SpCas9 is known to have high off-target editing rates natively. I would suggest that the authors compare editing vs. a high fidelity version of SpCas9 (SpCas9-HF1, HiFi-Cas9, or eSpCas9) on two additional sites to provide a baseline for specificity vs. these higher fidelity variants. I would expect that FrCas9 would have the highest on:off ratio against these variants.
4. The authors present compelling data on TATA-box editing, which is the most promising editing route for FrCas9, due to its 5'-NNTA-3' PAM sequence. As the TATA box is upstream of the gene to be regulated, I suggest that the authors design sgRNAs upstream of the TATA box and demonstrate reduction of expression using CRISPRa rather than FrCas9 nuclease. A dFrCas9-VPR or -VP64 construct would be suffice. If FrCas9 CRISPRa activates expression better than SpCas9 CRISPRa, due to its optimal positioning at the TATA box, this would be a very compelling argument to use FrCas9 for CRISPRa/i screening.

Minor Concerns

1. I suggest that the authors use a professional writing service to edit the grammar and spelling, as there are numerous errors that need to be fixed before the manuscript can be accepted.

2. The authors should upload their formatted bioinformatics code to GitHub for the reviewers. Otherwise, it is difficult to evaluate the identification pipeline.

Reviewer #2:

Remarks to the Author:

Cui et al. describe FrCas9, a previously uncharacterized Cas9 ortholog that recognizes a 5'-TA-3' PAM and that is active in genome editing in mammalian cells. In combination with other recent Cas9 editors that recognize different (CC, AA) two-nucleotide PAMs besides the GG of SpyCas9, this report provides a useful addition to the current roster of Cas9s. Importantly, they show that this platform can support base editing, where PAM availability (in a narrow window relative to the editing site) is crucial. This report could therefore be a valuable contribution to the field, though there are a number of problems with the draft that would need to be addressed first.

Although there are few addressable issues with the experiments (more on that below), in general the authors present a decent case that the PAM is indeed NNTA, that the activity in mammalian cells is strong, and that accuracy is sufficient for most editing purposes. However, the biggest problem by far is that they then take things too far and commit an unforced error, namely arguing unequivocally that this platform is superior to SpyCas9 in both activity and specificity. FrCas9 will be a very useful enzyme if it is as good, or even somewhat less good, than SpyCas9 in these respects, so (in my view) acceptance of this manuscript should not depend upon superiority over SpyCas9. The problem is that these claims of higher efficiency and accuracy would need to be backed by considerably more and better evidence than the authors provide. The numbers of guides and sites (with GUIDE-seq analyses) would need to be increased tremendously to get a reliable statistical sampling in support of these claims. The evidence would also need to go well beyond plasmid transfections in two transformed cell lines. We have no idea if any apparent activity/accuracy differences at any particular sites have to do with efficiency of transcription, translation, nuclear import, guide folding, guide loading, protein turnover, etc. etc. etc., rather than true efficiency differences. These other factors could exhibit cell-type-specific or cell-state-specific differences, and therefore preliminary indications of efficiency and accuracy may or may not pan out more broadly. Biochemical data would be crucial in support of the authors' claims, especially with respect to enzymatic efficiency, and there is essentially no biochemical analysis in this work. It would also be very helpful to know if the explicitly comparative experiments were done blind, as they should be. Overall, the manuscript would be better if the authors simply describe these preliminary indications of accuracy and efficiency as comparable to those of SpyCas9, pending deeper analyses at much larger numbers of sites.

Other concerns:

1. Lines 50-54: The authors invoke computational target site selections and protein engineering as the two strategies employed thus far to improve editing accuracy. They need to add guide engineering as a third, given previous reports about truncated guides (PMID 24463574), extended guides (24253446), and chemically modified guides (29377001, 29377001).
2. Lines 55-62: the claim that Cas9s other than SpyCas9s suffer from long PAMs was true once but is increasingly untenable. It is becoming a bit of a "straw man" argument and in this case is accompanied by cherry-picked examples and citations that omit recent Cas9s with two-nucleotide PAMs.
3. Line 80-81: "not close" should be defined more quantitatively and rigorously.
4. Lines 162-164: "directly compare" is misleading and not really true, because the guide sequences themselves are offset by one nucleotide. This is not as minor a change as it might sound because it could have substantial effects on guide folding and loading.
5. Lines 248-250: it makes no sense to invoke hydrogen bond stability in this context because PAMs are not unwound for recognition. How H-bond pairing by PAM residues could affect specificity is not explained, and such an attempt at explaining that claim would likely fail.
6. Lines 250-252: it is incorrect to state that longer guides necessarily imply or explain greater accuracy. See 1871108 (which is about ribozymes but uses logic that holds with other systems

that use pairing for recognition) and 28125791.

REVIEWER COMMENTS

Reviewer #1 (Remarks to the Author):

Major Concern 1. The authors utilize a depletion assay to assess PAM specificity. However, from the text and figures, it is not clear about the details of the assay and whether it would faithfully recapitulate the PAM sequences of other known Cas9 effectors. In addition to providing more detailed text within the manuscript, I would ask the authors to conduct an additional PAM determination assay (PAMDA) that has recently been published and evaluate PAM sequences in a more relevant context: <https://www.nature.com/articles/s41596-020-00465-2>. I also ask that they conduct comparisons with a few other Cas9s of their choosing for validation.

Response: Thank you for the helpful comment. According to your advice, we purchased the p11-Cas9_random_PAM-site1 plasmid (Addgene #160132) and pCMV-T7-SpCas9-P2A-EGFP plasmid (Addgene #139987). Then, we constructed the pCMV-T7-FrCas9-P2A-EGFP plasmid and synthesized the in vitro transcription templates of SpCas9 and FrCas9 sgRNAs for PAM-site1. Next, we conducted the HT-PAMDA experiments with timepoints including 1 min, 8 min and 32 min. We generated high-throughput data with sequencing depth of ~15,000,000 reads per timepoint, assuring sufficient coverage to resolve up to 5 nt of the PAM preference. To better visualize and understand the PAM preference from the HT-PAMDA data, we used the log-scale heatmap representing the absolute activity of 256 PAM sequences for the 1st - 4th PAM positions of SpCas9 and 2nd - 5th PAM positions of FrCas9 (Supplementary Fig. 2). The SpCas9 showed the PAM activities in the following

ranking of $NGG > NAG > NGA$, accompanied with the shifted NNGG PAMs, which were consistent with the results in the published paper¹. For FrCas9, we observed the canonical NNTA PAMs and the capacity to target sites with NNTG, NNAN and NNGT, which could also be reflected in the PAM weblogo from our depletion assay (Fig. 2e). We have added the HT-PAMDA results in Supplementary Fig. 2 and revised our manuscript correspondingly (Page 6, Line 114-117; Page 25-26, Line 534-543).

Supplementary Fig. 2. The PAM preference from HT-PAMDA experiments. The representation PAM requirements of WT-SpCas9 (a) and FrCas9 (b) using heatmaps. The heatmap was generated from the same HT-PAMDA data with two replicates per nuclease.

Major Concern 2. To confirm that editing rates on 5'-NNTA-3' PAM sequences in human cells are robust, I would suggest the authors do characterization on at least 24 targets in 8 different gene loci. Most papers do 48+ targets to show relevant editing, but as the authors demonstrate editing in other contexts, 24 different targets should suffice.

Response: Thank you for the constructive comment. Based on your suggestion, we have involved 32 sgRNAs targeting 8 different genes (each gene having 4 sites), including *ABCA1*, *ANAPC15*, *CDKN2A*, *DNMT1*, *EMX1*, *FANCF*, *RANKL* and *RUNX1*.

The TIDE assay showed that FrCas9 could generate robust cleavage in human cells under various sequence contents with 5'-NNTA-3' PAM in the above 32 loci (Supplementary Fig. 3). We have added this result in Supplementary Fig. 3 and revised our manuscript correspondingly (Page 6, Line 122-125).

Supplementary Fig. 3. The robust genome editing activity of FrCas9 on 5'-NNTA-3' PAM sequences in human cells. (a) The indel rates induced by FrCas9 in 8 human genes by 32 sgRNAs, which differed in the 2nd PAM base (5'-NATA-3' in green, 5'-NGTA-3' in yellow, 5'-NCTA-3' in blue, 5'-NTTA-3' in red). The assay was generated by TIDE in HEK293T cell line. Error bar indicated S.D. (n = 2).

Major Concern 3a. The authors use GUIDE-Seq to evaluate on- and off-target

efficiency. GUIDE-Seq is an internally controlled assay, not one that can be compared across samples. Thus, I would advise the authors to not use GUIDE-Seq to determine editing rate on target, but rather to calculate the ratio of on:off target reads as a measure of specificity at each site. Plotting these values across the different sites will provide a fuller understanding of FrCas9's specificity vs. SpCas9.

Response: Thank you for the valuable advice. We plotted the ratio of on:off target reads of FrCas9 and SpCas9 on 11 sites based on GUIDE-seq data in HEK293T (Fig 3e) and U2OS (Supplementary Fig. 8d) cell lines. As expected, we observed a significant higher on:off ratio in FrCas9 than SpCas9 in either HEK293T or U2OS cell lines ($P < 0.05$, paired *Wilcox* test). We have added the above data in Fig. 3e and Supplementary Fig. 8, and revised our manuscript correspondingly (Page 8-9, Line 173-178).

Fig. 3e and Supplementary Fig. 8d. The on:off ratio of GUIDE-seq reads of FrCas9 and SpCas9 at 11 sites in HEK293T (Fig.3e) and U2OS cell lines (Supplementary Fig. 8d).

Major Concern 3b. Furthermore, SpCas9 is known to have high off-target editing rates natively. I would suggest that the authors compare editing vs. a high-fidelity version of SpCas9 (SpCas9-HF1, HiFi-Cas9, or eSpCas9) on two additional sites to provide a baseline for specificity vs. these higher fidelity variants. I would expect that FrCas9 would have the highest on:off ratio against these variants.

Response: Thank you for the constructive comment. Based on your advice, we applied GUIDE-seq to evaluate the specificity of FrCas9, SpCas9, SpCas9-HF1, HiFi-Cas9, and eSpCas9 in *DYRK1A*-T2 and *GRIN2B*-T9 sites (top two active sites in the original manuscript) in HEK293T cell line. In *DYRK1A*-T2 site, the off-targets of each variant were FrCas9 (0), SpCas9 (15), SpCas9-HF1 (2), HiFi-Cas9 (2) and eSpCas9 (1). The off-targets in *GRIN2B*-T9 site were as below: FrCas9 (0), SpCas9 (8), SpCas9-HF1 (6), HiFi-Cas9 (12) and eSpCas9 (7) (Fig. 3f-g). As expected, the FrCas9 exhibited the highest on:off ratio in both sites (Fig. 3h). We have added the above results in Fig. 3f-h and revised our manuscript correspondingly (Page 9, Line 180-186).

Fig. 3. The genome-wide specificities of FrCas9 and SpCas9. (f-g) The off-targets of FrCas9, SpCas9, SpCas9-HF1, HiFi-Cas9 and eSpCas9 in *DYRK1A*-T2 (**f**) and *GRIN2B*-T9 (**g**) detected by GUIDE-seq in HEK293T cell line. (**h**) The on:off ratio of FrCas9, SpCas9, SpCas9-HF1, HiFi-Cas9, and eSpCas9 in *GRIN2B*-T9 and *DYRK1A*-T2 sites.

FrCas9, SpCas9, SpCas9-HF1, HiFi-Cas9 and eSpCas9 in *DYRK1A*-T2 and *GRIN2B*-T9. The ratio is defined by GUIDE-seq on-target reads dividing total off-target reads.

Major Concern 4. The authors present compelling data on TATA-box editing, which is the most promising editing route for FrCas9, due to its 5'-NNTA-3' PAM sequence. As the TATA box is upstream of the gene to be regulated, I suggest that the authors design sgRNAs upstream of the TATA box and demonstrate reduction of expression using CRISPRa rather than FrCas9 nuclease. A dFrCas9-VPR or -VP64 construct would be suffice. If FrCas9 CRISPRa activates expression better than SpCas9 CRISPRa, due to its optimal positioning at the TATA box, this would be a very compelling argument to use FrCas9 for CRISPRa/i screening.

Response: Thank you for the helpful comment. According to your advice, we tested FrCas9 CRISPRa using dFrCas9-VP64 directly targeting the TATA-box and compared its performance with dSpCas9-VP64 targeting the upstream of TATA-box. These experiments were conducted in *ABCA1*, *SOD1*, *GHI* and *BLM2* genes in HEK293T cell line. The results showed that dFrCas9-VP64 enabled effective transcriptional activation. Moreover, the fold activation of dFrCas9-VP64 in *ABCA1*, *GHI* and *BLM2* was higher than that of dSpCas9-VP64, while the fold activation of *SOD1* gene was comparable to that of dSpCas9-VP64 (Fig. 5g). Therefore, FrCas9 is a promising tool for CRISPR screening due to its unique 5'-NNTA-3' PAM. We added these results in Fig 5g and revised our manuscript correspondingly (Page 11,

Line 241-248).

Fig. 5g. The comparison of SpCas9 and FrCas9 CRISPRa in four genes with TATA-box. The fold activation of dSpCas9-VP64 and dFrCas9-VP64 in *ABCA1*, *SOD1*, *GH1* and *BLM2* genes was quantified by RT-qPCR. The experiments were conducted in HEK293T cells. Error bars indicated S.D. (n = 3 per group, * $P < 0.05$, ** $P < 0.01$, *** $P < 0.001$, **** $P < 0.001$, Student's t-test).

Minor Concern 1. I suggest that the authors use a professional writing service to edit the grammar and spelling, as there are numerous errors that need to be fixed before the manuscript can be accepted.

Response: Thank you for the kind advice. We have carefully checked the grammar and spelling errors in our manuscript and have corrected them one by one. The corresponding changes are made as following:

Page 1, Line 30: palindrome into palindromic

Page 2, Line 35: application into applications

Page 2, Line 42: tools into tool

Page 2, Line 46: generate into generates; occurred into occur

Page 2, Line 51: functional domain into functional domains

Page 3, Line 56: types into orthologs

Page 3, Line 65: system into systems

Page 3, Line 66: sequence into sequences

Page 4, Line 88: spacer and direct repeat into spacers and direct repeats

Page 7, Line 143: site into sites

Page 10, Line 215: windows into window

Page 22, Line 471: assemble into assembly

Page 35, Line 678: ration into ratio

Minor Concern 2. The authors should upload their formatted bioinformatics code to GitHub for the reviewers. Otherwise, it is difficult to evaluate the identification pipeline.

Response: Thank you for the helpful suggestion. We have uploaded our identification pipeline in GitHub (<https://github.com/Freya-Cui-2020/FrCas9>). Accordingly, we have added the related information in our revised manuscript (Page 30, Line 642-645).

Reviewer #2 (Remarks to the Author):

Major Concern 1. Cui et al. describe FrCas9, a previously uncharacterized Cas9 ortholog that recognizes a 5'-TA-3' PAM and that is active in genome editing in mammalian cells. In combination with other recent Cas9 editors that recognize different (CC, AA) two-nucleotide PAMs besides the GG of SpyCas9, this report

provides a useful addition to the current roster of Cas9s. Importantly, they show that this platform can support base editing, where PAM availability (in a narrow window relative to the editing site) is crucial. This report could therefore be a valuable contribution to the field, though there are a number of problems with the draft that would need to be addressed first.

Although there are few addressable issues with the experiments (more on that below), in general the authors present a decent case that the PAM is indeed NNTA, that the activity in mammalian cells is strong, and that accuracy is sufficient for most editing purposes. However, the biggest problem by far is that they then take things too far and commit an unforced error, namely arguing unequivocally that this platform is superior to SpyCas9 in both activity and specificity. FrCas9 will be a very useful enzyme if it is as good, or even somewhat less good, than SpyCas9 in these respects, so (in my view) acceptance of this manuscript should not depend upon superiority over SpyCas9. The problem is that these claims of higher efficiency and accuracy would need to be backed by considerably more and better evidence than the authors provide. The numbers of guides and sites (with GUIDE-seq analyses) would need to be increased tremendously to get a reliable statistical sampling in support of these claims. The evidence would also need to go well beyond plasmid transfections in two transformed cell lines. We have no idea if any apparent activity/accuracy differences at any particular sites have to do with efficiency of transcription, translation, nuclear import, guide folding, guide loading, protein turnover, etc. etc. etc., rather than true efficiency differences. These other factors could exhibit cell-type-specific or

cell-state-specific differences, and therefore preliminary indications of efficiency and accuracy may or may not pan out more broadly. Biochemical data would be crucial in support of the authors' claims, especially with respect to enzymatic efficiency, and there is essentially no biochemical analysis in this work. It would also be very helpful to know if the explicitly comparative experiments were done blind, as they should be. Overall, the manuscript would be better if the authors simply describe these preliminary indications of accuracy and efficiency as comparable to those of SpyCas9, pending deeper analyses at much larger numbers of sites.

Response: Thank you very much for the constructive suggestion. Indeed, we agreed that the data in our study was limited to draw such a claim. Therefore, we revised the manuscript according to your advice:

Title: "A new CRISPR/Cas9 system with higher editing efficiency and fidelity compared to SpCas9" was changed into "A new CRISPR/Cas9 system with high editing efficiency and fidelity".

Page 4, Line 69-70: "Further, we showed that FrCas9 achieved more efficient and safer genome editing than SpCas9 in human and HPV genome" was changed into "Further, we showed that FrCas9 could achieve comparable efficiency and specificity to SpCas9".

Page 10, Line 200-201: "FrCas9 exhibited superior efficiency and specificity than SpCas9 as an antiviral therapeutic tool" was changed into "FrCas9 exhibited remarkable efficiency and specificity as an antiviral therapeutic tool".

Page 12, Line 253-256: "Compared with the widely-used SpCas9, FrCas9 had higher

efficiency and specificity with the simple 2-nucleotide PAM sequences (5'-NNTA-3'). Therefore, FrCas9 may be more suited for genome engineering in mammalian cells than SpCas9 and will be a powerful and safe platform for biotechnological and clinical research” was changed into “FrCas9 hold high efficiency and specificity with simple 2-nucleotide PAM sequences (5'-NNTA-3'). Therefore, FrCas9 will be a powerful and safe platform for biotechnological and clinical research”.

Page 12, Line 276-277: “Taken together, FrCas9 with unique PAM (5'-NNTA-3') sequences has superior efficiency and specificity than widely used SpCas9” was changed into “Taken together, FrCas9 with unique PAM (5'-NNTA-3') sequence has high efficiency and specificity”.

Minor Concern 1. Lines 50-54: The authors invoke computational target site selections and protein engineering as the two strategies employed thus far to improve editing accuracy. They need to add guide engineering as a third, given previous reports about truncated guides (PMID 24463574), extended guides (24253446), and chemically modified guides (29377001).

Response: Thank you for the helpful comment. As you suggested, we added the guide engineering as the third strategy in the introduction section. The manuscript was revised as below (Page 3, Line 52-54): “The third is to engineer guide RNA, including truncated gRNAs², extended gRNAs³ and chemically modified gRNAs⁴”.

Minor Concern 2. Lines 55-62: the claim that Cas9s other than SpyCas9s suffer from

long PAMs was true once but is increasingly untenable. It is becoming a bit of a “straw man” argument and in this case is accompanied by cherry-picked examples and citations that omit recent Cas9s with two-nucleotide PAMs.

Response: Thank you for the kind advice. We agreed that we missed some Cas9s with two-nucleotide PAM, such as Nme2Cas9 (N4CC PAM)⁵ and SauriCas9 (NNGG PAM)⁶. We deleted the statement of “However, most above CRISPR effectors require longer PAM sequences (5'-NNGRRT-3' for SaCas9, 5'-NNNNGNTT-3' for NmeCas9, 5'-NNNVRYAC-3' for CjCas9, 5'-NNNNCRAA-3' for GeoCas9, 5'-NNNNRT-3' for PpCas9), which narrows down the targeting scopes” in our revised manuscript (Page 3, Line 59).

Minor Concern 3. Line 80-81: “not close” should be defined more quantitatively and rigorously.

Response: Thank you for the constructive comment. Based on your suggestion, we revised the sentence into “The phylogenetic analysis shows that FrCas9 is dissimilar to SpCas9 at a distance of 1.80 (Fig. 1a and Supplementary Table 1)” (Page 4, Line 76-77).

Minor Concern 4. Lines 162-164: “directly compare” is misleading and not really true, because the guide sequences themselves are offset by one nucleotide. This is not as minor a change as it might sound because it could have substantial effects on guide folding and loading.

Response: Thank you for the helpful suggestion. We have changed “we could directly compare their genome editing efficiency and specificity in sequence with 5'-GGTA-3'.” into “we compared their genome editing efficiency and specificity in sequence with 5'-GGTA-3'” (Page 8, Line 165-166).

Minor Concern 5. Lines 248-250: it makes no sense to invoke hydrogen bond stability in this context because PAMs are not unwound for recognition. How H-bond pairing by PAM residues could affect specificity is not explained, and such an attempt at explaining that claim would likely fail. Lines 250-252: it is incorrect to state that longer guides necessarily imply or explain greater accuracy. See 1871108 (which is about ribozymes but uses logic that holds with other systems that use pairing for recognition) and 28125791.

Response: Thank you for the kind comment. We agreed with your comment and deleted this context in our revised manuscript (Page 12, Line 257).

References

- 1 Walton, R. T., Hsu, J. Y., Joung, J. K. & Kleinstiver, B. P. Scalable characterization of the PAM requirements of CRISPR-Cas enzymes using HT-PAMDA. *Nat Protoc* **16**, 1511-1547, doi:10.1038/s41596-020-00465-2 (2021).
- 2 Fu, Y., Sander, J. D., Reyon, D., Cascio, V. M. & Joung, J. K. Improving CRISPR-Cas nuclease specificity using truncated guide RNAs. *Nat Biotechnol* **32**, 279-284, doi:10.1038/nbt.2808 (2014).
- 3 Cho, S. W. *et al.* Analysis of off-target effects of CRISPR/Cas-derived RNA-guided

endonucleases and nickases. *Genome Res* **24**, 132-141, doi:10.1101/gr.162339.113

(2014).

- 4 Yin, H. *et al.* Partial DNA-guided Cas9 enables genome editing with reduced off-target activity. *Nat Chem Biol* **14**, 311-316, doi:10.1038/nchembio.2559 (2018).
- 5 Edraki, A. *et al.* A Compact, High-Accuracy Cas9 with a Dinucleotide PAM for In Vivo Genome Editing. *Mol Cell* **73**, 714-726 e714, doi:10.1016/j.molcel.2018.12.003 (2019).
- 6 Hu, Z. *et al.* A compact Cas9 ortholog from *Staphylococcus Auricularis* (SauriCas9) expands the DNA targeting scope. *PLoS Biol* **18**, e3000686, doi:10.1371/journal.pbio.3000686 (2020).

Reviewers' Comments:

Reviewer #1:

Remarks to the Author:

In this revision, Cui, et al. supports their argument that their novel Cas9 from *Faecalibaculum rodentium* simultaneously possesses broad 5'-NNTA-3' PAM specificity, efficient editing and regulation in human cells, and high specificity in relation to off-targets. They conduct a thorough characterization of their enzyme using bioinformatic analyses, PAM specificity analysis via PAMDA, GUIDE-Seq, and CRISPRa assays. Furthermore, they engineer base editors incorporating FrCas9 and show high base editing efficiency. Most interestingly, FrCas9 maintains a palindromic PAM, enabling targeting of pairwise targets in opposite orientations, opening up a more significant percentage of genomes as compared to SpCas9.

Overall, the authors do an excellent job of responding to reviewer comments and have presented a version of their manuscript that is suitable for acceptance and publication in *Nature Communications*. I am overall very excited to see the enzyme be utilized for various applications, from editing to genome screening.

Reviewer #2:

Remarks to the Author:

The authors responded constructively to the critiques and the paper is significantly improved.

REVIEWERS' COMMENTS

Reviewer #1 (Remarks to the Author):

In this revision, Cui, et al. supports their argument that their novel Cas9 from *Faecalibaculum rodentium* simultaneously possesses broad 5'-NNTA-3' PAM specificity, efficient editing and regulation in human cells, and high specificity in relation to off-targets. They conduct a thorough characterization of their enzyme using bioinformatic analyses, PAM specificity analysis via PAMDA, GUIDE-Seq, and CRISPRa assays. Furthermore, they engineer base editors incorporating FrCas9 and show high base editing efficiency. Most interestingly, FrCas9 maintains a palindromic PAM, enabling targeting of pairwise targets in opposite orientations, opening up a more significant percentage of genomes as compared to SpCas9.

Overall, the authors do an excellent job of responding to reviewer comments and have presented a version of their manuscript that is suitable for acceptance and publication in Nature Communications. I am overall very excited to see the enzyme be utilized for various applications, from editing to genome screening.

Response: Thank you for the positive comments.

Reviewer #2 (Remarks to the Author):

The authors responded constructively to the critiques and the paper is significantly improved.

Response: Thank you for the kind comments.